**Data Availability Statement:** All relevant data are within the manuscript and its Supporting Information files.

**Funding:** National Natural Science Foundation of China (Grant Number: 71974091) Philosophy and

# Multi-group symbiotic evolutionary mechanisms of a digital innovation ecosystem: Numerical simulation and case study

**Yuqiong Li, Liping Wu** ®*

School of Economics Management and Law, University of South China, Hengyang, P. R. China

* 19907344375@163.com

## Abstract

In the digital innovation ecosystem, the symbiosis mode formed between ecosystem members not only relates to their survival and development but also affects the ecosystem's symbiosis evolution mechanism. Based on symbiosis theory, this study first explores the evolutionary equilibrium strategy and its stability for three types of populations—core enterprises, digital platforms, and university research institutes—and then uses numerical simulation and a case study to explore the symbiotic evolution mechanism of the digital innovation ecosystem. The results show that: First, the digital innovation ecosystem is a complex adaptive system in which the three types of populations form different symbiotic relationships under different symbiotic modes and conduct symbiotic activities, such as value co-creation, to characterize the unique symbiotic evolutionary structure. Second, in this ecosystem, the symbiotic relationship formed by the combined values of different symbiotic coefficients between populations determines the outcome of symbiotic evolution. Third, the ideal direction of the evolution of the digital innovation ecosystem is a mutually beneficial symbiotic relationship. Thus, the symbiotic relationship between populations should be transformed into a mutually beneficial symbiotic relationships as much as possible. This study makes theoretical contributions by shedding light on the symbiotic evolution mechanism of the digital innovation ecosystem. It also offers countermeasures for the digital innovation cooperation of various stakeholders in China's digital innovation ecosystem.

## Introduction

The digital economy has become a new driving force for economic development, disrupting the ways in which enterprises gain sustainable competitive advantages [1]. In an era in which everything can be digitized, the value co-creation among innovation actors has also been reshaped. Today, digital elements are being continuously integrated into the operation and development of innovation ecosystems. This has inspired academic discussions on the construction of digital innovation ecosystems [2,3]. Building digital innovation ecosystems is

Social Science Foundation of Hunan Province (Grant Number: 21JD023) Natural Science Foundation of Hunan Province (Grant Number: 2023JJ50134) The funders had no role in study design, data collection and analysis, decision to publish, or preparation of the manuscript.

**Competing interests:** There are no conflicts of interest disclosed in the submission of this article, and the article is approved for publication by all authors.

essential for breaking down the boundaries of different industries, regions, and enterprises, and realizing digital value co-creation [4]. Thus, establishing digital innovation ecosystems is both academically and practically important.

Numerous studies have examined the notion of the digital innovation ecosystem, its progress, associated value generation, and the game relationship of each stakeholder. Chae observed that the digital innovation ecosystem consists of digital innovation and innovation ecosystems [5]. It is a complex system of competition symbiosis in which different innovation actors and environments in a particular region combine various digital and non-digital resources using digital technology to create new products and services [6]. Kolloch and Dellermann pointed out that the rise of the Internet of Things has brought ecosystems to the forefront of innovation, and effectively managing the evolution of these ecosystems during the process of digital innovation is important [7]. Suseno et al. examined how value is generated through the interaction between consumers and professional stakeholders in a digital innovation ecosystem [8]. Liu et al. proposed a tripartite evolutionary game model of digital transformation based on expectation theory and Lyapunov's first law to analyze the obstacles faced by stakeholders in the digital ecosystem for undertaking digital transformation [6].

With the development of the digital economy, the digital innovation ecosystem can be characterized by new symbiotic structures, such as the fluidization of data elements, empowerment of digital technologies, and organization of digital platforms [5,9,10]. Furthermore, except for the digital ecosystems of giants such as Apple and Google, both of which have seen unprecedented innovation wealth amassed by the inclusion of numerous stakeholders, most other digital innovation ecosystems are in dire straits [11]. Symbiosis theory provides a theoretical reference for the development and growth of innovation ecosystems [12]. Exploring the symbiotic evolution mechanism of digital innovation ecosystems can promote synergistic coexistence among various stakeholders to achieve stable development [13]. Nevertheless, the aforementioned research concentrates primarily on the evolutionary relationships and linkages of digital innovation ecosystems, while the mutual evolutionary mechanism of digital innovation ecosystems remains unexplored. Only a few studies examine the symbiotic development of digital innovation ecosystems and cannot provide mature guidance for digital innovation practices in China.

This study examines the symbiotic evolutionary mechanisms of digital innovation ecosystems and introduces the crowd growth model Logistic equation in ecological theory to explore the evolution patterns of three types of symbiotic units–core enterprises, digital platforms, and university research institutes–in other symbiotic relationships within the digital innovation ecosystem. Then, by integrating numerical simulations and conducting a case study, we try to understand the mechanism and trajectory of development that leads to the mutual enhancement of the Chinese digital innovation and ecological system. This study offers a theoretical explanation and policy insights into digital innovation practices. The primary research questions are as follows:

**RQ1:** How have these three populations evolved over time in different symbiotic relationships?

**RQ2:** What key factors affect the symbiotic evolutionary equilibrium of digital innovation ecosystems?

**RQ3:** How can we optimize the symbiotic evolution mechanism of the digital innovation ecosystem?

The essay is organized as follows. The literature review and theoretical framework section discusses the relevant literature and establishes our theoretical framework. The research and data methodology section constructs a symbiotic evolutionary model, and performs the numerical simulations and case analysis. The discussion section discusses the numerical

simulation results and analyzes their practical significance in the context of real-world cases. Finally, the conclusion section summarizes the conclusions and implications.

## Literature review and theoretical framework

### Literature review

**Innovation ecosystem evolution.**  The concept of creative ecosystems is derived from ecology and represents a harmonious fusion of ecology, systems science, and innovation theory [4]. The economist initially introduced the notion of innovation [14], followed by Tansley's concept of ecosystems [15]. Moore originally developed the idea of an enterprise ecosystem by comparing biological systems and enterprise competition, and defined it as a state of economic coherence that arises from organizational interaction [16]. Subsequently, the United States Presidential Council of Scientific and Technological Advisers introduced the concept of an "innovative ecosystem" for the first time in 2004 [17]. This concept is based on the idea that the innovation paradigm has transitioned into an innovative ecosystem phase, which is referred to as Innovation Paradigm 3.0 [18].

The innovation ecosystem is an intricate system that involves interdependent relationships, competition, and collaboration among various entities involved in innovation. It exhibits nonlinear interactions and complex network structures [19]. Notably, the study of the innovation ecosystem's evolution process has consistently been a significant topic in innovation theory-related research. Li et al. pointed out that the innovation ecosystem members are connected in integrating innovation resources, which pushes them to shift from independent development to symbiotic evolution [20]. Ander proposed that the core enterprise primarily controls the innovation ecosystem, which collaborates with supporting organizations to produce mutual benefits and drive system progress. The author also emphasized that the innovation ecosystem is predominantly propelled by the central company. Thus, by promoting cooperation between the leading enterprise and supporting organizations, a mutually advantageous result can be achieved, resulting in the development of the system [21]. According to Still et al., the fundamental aspect of the evolution of the innovation ecosystem is the symbiotic evolutionary interaction between key members [22]. Several scholars have extensively examined the development of innovation ecosystems from various research angles and methodologies, focusing on distinct core firms as primary entities [23–26] and the elements that influence this evolution [27,28].

**Digital innovation ecosystem evolution.**  With the advent of the Digital Economy 3.0 era, digital elements have entered the innovation ecosystem, giving new digital identities to the original innovation subjects. The swift and extensive digitization of innovation processes and outcomes has questioned established innovation management theories. Many scholars contend that owing to the impact of digitization on the innovation process, novel approaches are needed in the current digital innovation and transformation era [29–31]. Consequently, the theory of digital innovation ecosystems has emerged.

The digital innovation ecosystem is an intricate organizational framework comprising core firms, digital platforms, and research institutions [32,33], collaborating to achieve their advantages [5]. According to symbiotic evolution theory, species can enhance the ongoing evolution of groups by trading resources with other relevant species, and building enduring and stable cooperative interactions [12]; these can ultimately lead to the formation of symbiotic links between populations. Chae underlined the significance of complex networks in offering a conceptual framework for examining the birth and development of digital innovation ecosystems, contributing conceptually and methodologically to understanding digital innovation ecosystems [5]. Zou et al. designed an evolutionary game model that incorporates core firms,

university research institutes, and information intermediaries as critical participants in the game, which entails advancing positive engagement and the mutually beneficial development of knowledge innovation capacity between digital innovation ecosystems and leading stakeholders in innovation [32]. Liu investigated the mechanism of knowledge transfer inside digital platforms in digital innovation ecosystems using modeling and simulation techniques. The author attempted to establish a theoretical foundation for digital innovation subjects within the system, enabling them to acquire digital value and strengthen their competitive advantage through knowledge sharing [33]. Beltagui et al. developed a comprehensive four-stage process model of digital innovation ecosystems using a case study approach to address the research gaps related to digital innovation ecosystems, particularly concerning mutational development and disruptive innovation [34]. Beliaeva et al. established the progression in the capacity of digital enterprises to position themselves within digital innovation ecosystems, evolving from a rudimentary state to a more dynamic one [35]. Li et al. explored the influence of various stakeholder engagement strategies on risk prevention within digital innovation ecosystems via the lens of risk evolution [36]. Increasingly, many companies have established digital innovation ecosystems to facilitate value exchange [37] and overcome the obstacles associated with digital technology innovation. Renowned corporations such as Hail, Facebook, Google, and Amazon have established collaborative digital innovation ecosystems to obtain competitive advantages [4]. While both scholars and practitioners recognize the importance and potential growth of the digital innovation ecosystem, research in this area is still in the early stages of exploration and development [38]. Significant potential exists for further advancement, particularly in the study of symbiotic evolution.

In summary, on the one hand, research on the evolution of innovation ecosystems has laid a solid theoretical foundation for further research on digital innovation ecosystems. On the other hand, research on digital innovation ecosystems' evolution mainly focuses on evolution paths, development, and countermeasures. There is significant potential for symbiotic evolution research. However, studies mostly use a single research method and lack a comprehensive analysis with multiple research methods.

To fill these gaps, this study makes the following contributions: (1) Based on symbiosis and ecology theories, this study constructs a symbiotic evolution model of the digital innovation ecosystem and introduces a Logistic growth model to analyze the symbiotic evolution law of the three types of populations in the ecosystem—core enterprises, digital platforms, and university research institutes. (2) Considering the symbiotic evolution mechanism of the digital innovation ecosystem as the research object, this study combines numerical simulation methods and actual Chinese cases of digital innovation ecosystems to understand this mechanism and obtain more generalizable insights. (3) This study comprehensively considers the influence of various symbiotic coefficients on the evolutionary equilibrium of the digital innovation ecosystem. It clarifies the different roles and values of the three types of populations in the digital innovation ecosystem, and explores the dynamic evolutionary process of these populations.

## Theoretical framework

Ecological and symbiotic theories lay the theoretical foundation for studying the symbiotic evolutionary mechanisms in digital innovation ecosystems [39,40]. The digital innovation ecosystem is defined by the virtualization of entities [41,42], indistinct borders of systems [43], and digitization of components [29]. Its symbiotic evolution is a transition process from the lower ecological level to the higher environmental level [23]. The evolutionary system of an innovation ecosystem comprises symbiotic units, patterns, and environments. The interaction

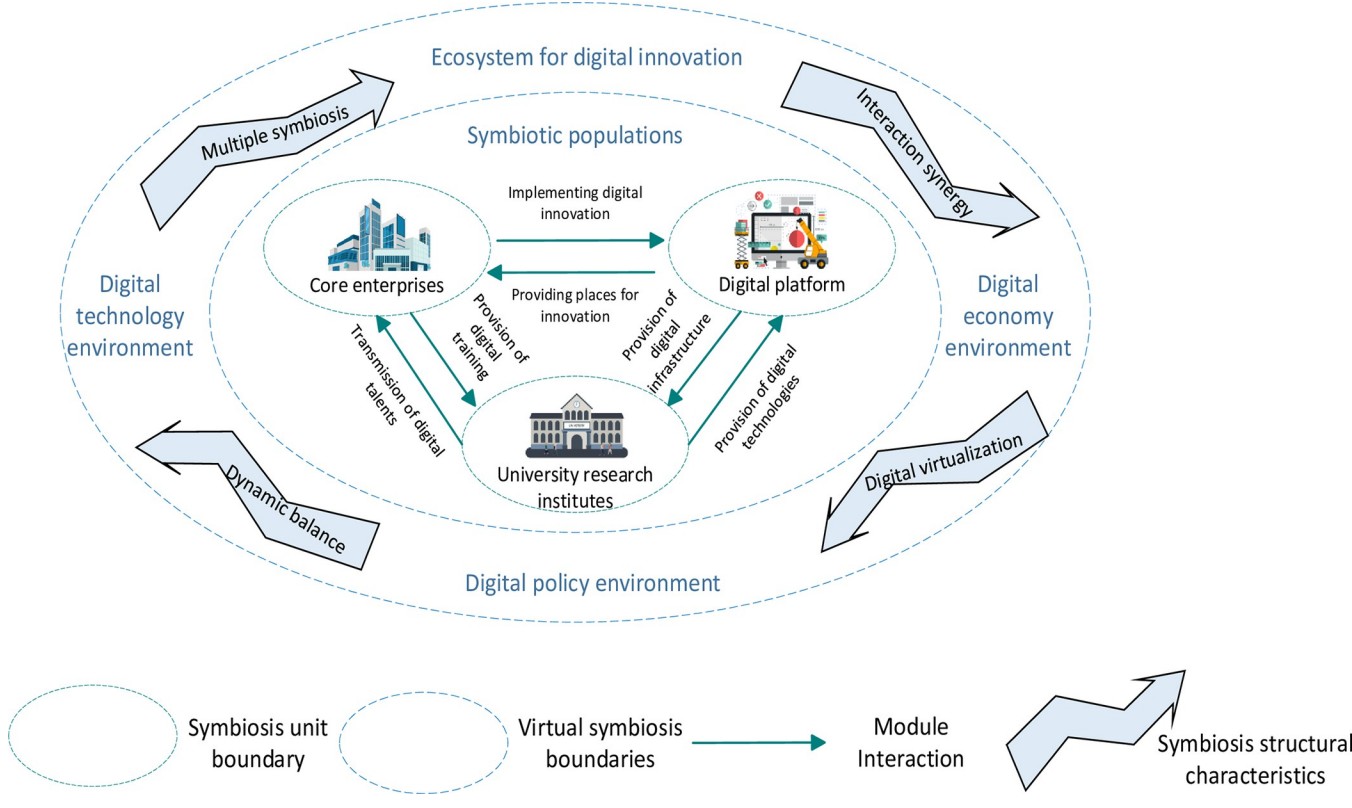

**Fig 1. A theoretical model for the symbiotic evolution of digital innovation ecosystems.**

between mutually dependent entities following a given path within a particular domain is called the symbiotic effect [40]. The symbiotic module is a fundamental component of the digital innovation ecosystem, and serves as the core unit for data generation and information exchange throughout the system. The digital innovation ecosystem comprises three main components: the core enterprise symbiotic module, which implements digital innovation; the digital platform symbiotics module, which provides digital infrastructure; and university research institutions, which contribute to digital R&D technology [44]. These components exist within a specific symbiotic environment and engage in various digital innovation interaction activities, such as value creation, within different symbiosis patterns. Symbiosis patterns are core elements of the digital innovation ecosystem and encompass five distinct types of synergistic relationships: freestanding, competitive, parasitic, biased mutual benefit, and mutually beneficial. A symbiotic environment refers to the external conditions under which the symbiosis model exists, including the ecological, technological, economic, and policy aspects of digital innovation, as depicted in Fig 1.

Within the digital innovation ecosystem, core enterprises hold a dominant position, whereas digital platforms and university scientific institutes play supporting roles by providing services and technologies to core enterprises. They collaborate to achieve value co-creation and enhance system synergies. As innovative entities, core enterprises can use digital platforms to implement digital innovation [32] and leverage existing outcomes to offer digital employment services to university research institutions, relying on data and information from digital platforms and university scientific institutions to enhance the system's overall value. As a combination of digital resources covering services and technologies, digital platforms provide a

virtual space for interaction and data sharing for innovative subjects, free from spatiotemporal constraints, and provide accurate digital services for value creation through digital technology connections between core enterprises and university research institutes [33,38,45–47]. As the primary entities involved in applying knowledge and developing new technologies, university research institutions can supply core enterprises with highly skilled digital technology professionals, enabling knowledge technology experts to transition from laboratories to actively contributing to enterprise operations. In addition, university research institutions provide the digital platform's necessary digital technology infrastructure and support, facilitating its continuous improvement and advancement [48].

The symbiosis mode refers to the coexistence mode that arises from the interaction of symbiotic units, serving as the fundamental essence of a digital innovation ecosystem. The symbiosis mode encompasses five distinct types as defined before. The freestanding symbiosis mode occurs when the three symbiotic units within a symbiotic population do not affect each other, and develop separately and autonomously. The competitive symbiosis mode occurs when symbiotic units within a population compete for the same digital innovation resources. Consequently, some symbiotic units die due to resource scarcity, while the surviving units continue to thrive and grow. The parasitic symbiosis represents the existence of a parasitic relationship between symbiotic units in a symbiotic population, in which the parasitic symbiotic unit benefits from the resources of the parasitized symbiotic unit, which in turn suffers a certain degree of damage. The biased symbiosis mode illustrates the appearance of preferential synergies among the symbiotic units in a symbiotic population. Specifically, the benefits of the symbioses do not impact the development of other symbiotes. Finally, the mutually beneficial mode symbolizes the most optimal form of synergy, where the symbiosis units within the symbiotic population have a mutually beneficial relationship, leading to mutual benefits and a win-win situation.

The symbiotic environment is the external environment in which a symbiotic mode operates. This environment essentially involves the digital innovation ecological, digital technology, digital economy, and digital policy environments. Generally speaking, the higher the ecological niche suitability of the digital innovation ecosystem, the more developed the digital technology, the more prosperous the digital economy, and the more perfect the digital policy, the higher the symbiosis level and tier of the symbiotic populations in the digital innovation ecosystem.

According to the symbiosis and ecology theories, and supported by existing research, the evolution of the digital innovation ecosystem exhibits the following structural characteristics. First, diversity symbiosis pertains to the distinctive attributes of the evolutionary patterns shown by different forms of symbiosis within the digital innovation ecosystem [49]. Interactive synergy refers to the rapid development of emerging information technology and the prominent appearance of digital platforms. Here, digital innovation subjects, such as core enterprises and university research institutes, can have more efficient cross-border interaction [50]. Digital modularity arises from digital technology, which enables the digital innovation ecosystem to exhibit a layered modular structure. By employing virtual reality technology, the digital space can be further expanded, thereby facilitating the broadening of the scope of digital innovation applications and generation of additional new technologies [34,51]. This, in turn, enhances the promotion of system synergy evolution. Dynamic balance emphasizes the evolution of the digital innovation system, showing the characteristics of a gradual sequence from nothing to nothing and accumulation to multiplicity [49]. Within the digital innovation ecosystem, symbiotic units engage in collaborative efforts for shared interests, interact within a dynamic environment, and exhibit many symbiotic, synergistic, and mutually advantageous synergies, ultimately contributing to ecological equilibrium.

## Research and data methodology

### Model construction

Zhang examined the evolutionary dynamics of new populations inside creative ecosystems. The author's conclusions support the argument that these dynamics adhere closely to the Logistic growth rules observed in ecological systems, and that digital innovation ecosystems share common characteristics and synergies with natural ecosystems [23]. Meanwhile, Yu et al. constructed a quantum evolutionary game model and combined it with a case study to investigate the mutually beneficial evolutionary stabilization strategies of major players in industrial complex systems [52]. Yin et al. investigated the stochastic differential game of low-carbon technology sharing in the collaborative innovation system of different types of enterprises, demonstrating that game theory can be useful for studying complex systems [53]. Building on this theoretical foundation, this study seeks to develop a Logistic population growth model that assumes a symbiotic dynamic evolution of three types of populations within the digital innovation ecosystem. The stability of the equilibrium points within the model are analyzed using game theory [54,55], and the evolutionary law for the various populations within the digital innovation ecosystem is derived.

**Hypothesis.** This research is grounded on the following hypotheses:

**Hypothesis 1:** Within the digital innovation ecosystem, core enterprises, digital platforms, and university research institutions are the main forces of digital innovation. The digital innovation eco-environment consists of core enterprise groups $C_i(i = 1, 2 \ldots l)$, digital platforms groups $D_j(j = 1, 2 \ldots m)$, and university research institutions categories $E_k(k = 1, 2 \ldots n)$. These three populations are limited by objective conditions, such as technology, resources, and institutions, and their growth process is consistent with the natural ecosystem according to the Logistic growth law: when the marginal output of one group is equal to the marginalized inputs, the population stops growing to its maximum size.

**Hypothesis 2:** In a digital innovation ecosystem, the magnitude of the growth process of the population of innovations indicates the caliber of its growth. A larger scale corresponds to a higher quality of growth within the innovative population, resulting in a greater allocation of the system's resources towards innovation and an enhanced capacity to generate better value.

**Hypothesis 3:** Many innovation populations interact within the digital innovation ecosystem. Remarkably, the core enterprise population has an overwhelming advantage. Furthermore, five distinct synergistic relationships exist among core enterprises, digital platform communities, and university research institutions. The evolutionary trajectory of an inventive population is influenced not only by shifts in its norms but also by changes in the scale of other innovative populations. This interaction exhibits a prominent limitation and bias, as the symbiotic component demonstrates weak intensity. Meanwhile, the reciprocal relationship between positively and negatively affected populations is compromised. The symbiotic factor reveals a negative (positive) impact on the beneficial (detrimental) population.

**Formulas.** Based on the above assumptions, a multi-population symbiotic evolutionary dynamic model for a digital innovation ecosystem is constructed as follows:

$$\begin{cases} \dfrac{dy_e}{dt} = r_e\left[1 - \dfrac{y_e}{N_e} - \alpha_{de}\dfrac{y_d}{N_d} - \alpha_{ce}\dfrac{y_c}{N_c}\right]y_e, y_e(0) = y_{e0} \\[2ex] \dfrac{dy_d}{dt} = r_d\left[1 - \dfrac{y_d}{N_d} - \alpha_{ed}\dfrac{y_e}{N_e} - \alpha_{cd}\dfrac{y_c}{N_c}\right]y_d, y_d(0) = y_{d0} \\[2ex] \dfrac{dy_c}{dt} = r_c\left[1 - \dfrac{y_c}{N_c} - \alpha_{ec}\dfrac{y_e}{N_e} - \alpha_{dc}\dfrac{y_d}{N_d}\right]y_c, y_c(0) = y_{c0} \end{cases} \quad (1)$$

**Table 1. Symbiotic model of digital innovation ecosystems.**

| Combination of values | Symbiotic evolutionary mode | Descriptions |
|---|---|---|
| $\alpha_{de}, \alpha_{ce}, \alpha_{ed}, \alpha_{cd}, \alpha_{ec}, \alpha_{dc} = 0$ | Freestanding symbiotic mode | Three categories of digital innovation clusters do not interfere with each other and develop independently. |
| $\alpha_{de}, \alpha_{ce}, \alpha_{ed}, \alpha_{cd}, \alpha_{ec}, \alpha_{dc} > 0$ | Competitive symbiosis mode | Three categories of digital innovation populations compete for innovative resources, when the symbiosis factor is less than 1, for the equal competition model; when the synergies factor is not equal and the arbitrary one-sided symbiosis factor is greater than 1, the malignant competitive model. |
| $\alpha_{de}\alpha_{ed} < 0;$<br>$\alpha_{ce}\alpha_{ec} < 0;$<br>$\alpha_{dc}\alpha_{cd} < 0;$ | Parasitic symbiotic mode | Populations with antagonistic symbiosis benefited, while populations with positive symbioses suffered. |
| $\alpha_{de} = 0, \alpha_{ed} < 0;$<br>$\alpha_{de} < 0, \alpha_{ed} = 0;$<br>$\alpha_{ce} = 0, \alpha_{ec} < 0;$<br>$\alpha_{ce} < 0, \alpha_{ec} = 0;$<br>$\alpha_{dc} = 0, \alpha_{cd} < 0;$<br>$\alpha_{dc} < 0, \alpha_{cd} = 0;$ | Biased mutual benefit mode | Populations exhibiting antagonistic symbiosis experience advantageous effects, while populations displaying zero symbiosis remain unaffected. |
| $\alpha_{de}, \alpha_{ce}, \alpha_{ed}, \alpha_{cd}, \alpha_{ec}, \alpha_{dc} < 0$ | Mutually beneficial symbiotic model | All three categories of digital innovation clusters will yield advantages. |

In this scenario, the variables $y_e(t), y_d(t),$ and $y_c(t)$ represent the population sizes of core enterprises, digital platforms, and university research institutes, respectively, within the digital innovation ecosystem. The parameter $t = 0$ signifies the original population size for all groups. Beyond that, the most significant number of population sizes for each group, subject to constraints imposed by constrained system resources, are designated as $N_e, N_d, N_c$. The growth rates of the core enterprise populations, digital platform populations, and university research institutes are denoted as $r_e, r_d,$ and $r_c$, respectively. $r_e y_e, r_d y_d,$ and $r_c y_c$ indicate the respective development trends of the corresponding populations. $1 - \frac{y_e}{N_e}, 1 - \frac{y_d}{N_d},$ and $1 - \frac{y_c}{N_c}$ are Logistic coefficients, implying that the three populations impacted by the restricted resources of the system are consumed by other populations and hinder their own growth on a broader scale. $\alpha_{ij}(i \neq j, i, j = e, d, c)$ express the coefficient of symbiosis of group i to group j. Referring to Modis [56], the type of interaction between populations can be determined based on various values of $\alpha$, as shown in Table 1.

**Stability analysis.** To explore the principles of symbiotic evolution and dynamic processes within different factions of the digital innovation ecosystem, Eq (1) of the multi-population symbiotic evolutionary dynamic equations of the digital innovation ecosystem is zero, as given in Eq (2) below.

$$\begin{cases} r_e\left[1 - \dfrac{y_e}{N_e} - \alpha_{de}\dfrac{y_d}{N_d} - \alpha_{ce}\dfrac{y_c}{N_c}\right]y_e = 0 \\ r_d\left[1 - \dfrac{y_d}{N_d} - \alpha_{ed}\dfrac{y_e}{N_e} - \alpha_{cd}\dfrac{y_c}{N_c}\right]y_d = 0 \\ r_c\left[1 - \dfrac{y_c}{N_c} - \alpha_{ec}\dfrac{y_e}{N_e} - \alpha_{dc}\dfrac{y_d}{N_d}\right]y_c = 0 \end{cases} \quad (2)$$

Eq (2) is used to derive eight local balancing points of the symbiotic evolution of the digital innovation ecosystem: $E_1(0,0,0), E_2(N_e,0,0), E_3(0, N_d,0), E_4(0,0, N_c), E_5$

$$\left(0, \frac{(\alpha_{cd}-1)N_d}{\alpha_{dc}\alpha_{cd}-1}, \frac{(\alpha_{dc}-1)N_c}{\alpha_{dc}\alpha_{cd}-1}\right), E_6\left(\frac{(\alpha_{ce}-1)N_e}{\alpha_{ec}\alpha_{ce}-1}, 0, \frac{(\alpha_{ec}-1)N_c}{\alpha_{ec}\alpha_{ce}-1}\right), E_7\left(\frac{(\alpha_{de}-1)N_e}{\alpha_{de}\alpha_{ed}-1}, \frac{(\alpha_{ed}-1)N_d}{\alpha_{de}\alpha_{ed}-1}, 0\right), E_8(y_e^*, y_d^*, y_c^*).$$

$$\begin{cases} y_e^* = \dfrac{(\alpha_{dc}\alpha_{cd} - \alpha_{dc}\alpha_{ce} - \alpha_{de}\alpha_{cd} + \alpha_{de} + \alpha_{ce} - 1)N_e}{\alpha_{ed}\alpha_{de} + \alpha_{ec}\alpha_{ce} + \alpha_{dc}\alpha_{cd} - \alpha_{ed}\alpha_{dc}\alpha_{ce} - \alpha_{ec}\alpha_{de}\alpha_{cd} - 1} \\[2mm] y_d^* = \dfrac{(\alpha_{ec}\alpha_{ce} - \alpha_{ec}\alpha_{cd} - \alpha_{ed}\alpha_{ce} + \alpha_{ed} + \alpha_{cd} - 1)N_d}{\alpha_{ed}\alpha_{de} + \alpha_{ec}\alpha_{ce} + \alpha_{dc}\alpha_{cd} - \alpha_{ed}\alpha_{dc}\alpha_{ce} - \alpha_{ec}\alpha_{de}\alpha_{cd} - 1} \\[2mm] y_c^* = \dfrac{(\alpha_{ed}\alpha_{de} - \alpha_{ed}\alpha_{dc} - \alpha_{ec}\alpha_{de} + \alpha_{ec} + \alpha_{dc} - 1)N_c}{\alpha_{ed}\alpha_{de} + \alpha_{ec}\alpha_{ce} + \alpha_{dc}\alpha_{cd} - \alpha_{ed}\alpha_{dc}\alpha_{ce} - \alpha_{ec}\alpha_{de}\alpha_{cd} - 1} \end{cases} \tag{3}$$

The Jacobite Matrix J of the dynamic evolutionary system is:

$$J = \begin{pmatrix} r_e\left(1 - \dfrac{2y_e}{N_e} - \alpha_{de}\dfrac{y_d}{N_d} - \alpha_{ce}\dfrac{y_c}{N_c}\right) & \dfrac{-r_e y_e \alpha_{de}}{N_d} & \dfrac{-r_e y_e \alpha_{ce}}{N_c} \\[3mm] \dfrac{-r_d y_d \alpha_{ed}}{N_e} & r_d\left(1 - \dfrac{2y_d}{N_d} - \alpha_{ed}\dfrac{y_e}{N_e} - \alpha_{cd}\dfrac{y_c}{N_c}\right) & \dfrac{-r_d y_d \alpha_{cd}}{N_c} \\[3mm] \dfrac{-r_c y_c \alpha_{ec}}{N_e} & \dfrac{-r_c y_c \alpha_{dc}}{N_d} & r_c\left(1 - \dfrac{2y_c}{N_c} - \alpha_{ec}\dfrac{y_e}{N_e} - \alpha_{dc}\dfrac{y_d}{N_d}\right) \end{pmatrix} \tag{4}$$

The Jacobi Matrix is a valuable tool for assessing the stability of a system's equilibrium point. According to the criteria outlined in prior research [57], a balance point can be classified as a progressive stability balance point if it simultaneously satisfies the conditions det(J)>0 and tr(J)<0. Table 2 presents a stability analysis of the symbiotic evolution of the digital innovation ecosystem balance points.

Notably, the symbiotic evolution of the digital innovation ecosystem has three stable points: $E_2(N_e,0,0)$, $E_3(0, N_d,0)$, $E_4(0,0, N_c)$. The value combination of the symbiotic factor plays a crucial role in affecting the stability of the balance point of the digital innovation ecosystem. When $\alpha_{ed} > 1, \alpha_{ec} > 1$, a competitive relationship exists between core enterprise and digital platform populations, as well as university research institution populations. The core enterprise populations have a significant competitive edge, surpassing the other two groups in terms of comparable digital innovation resources. Consequently, the other two groups are projected to decline because of the scarce supply of resources, whereas the core business

**Table 2. Stability analysis of symbiotic evolution of digital innovation ecosystems.**

| Equilibrium point | det(J) | tr(J) | Stability conditions | ESS |
|---|---|---|---|---|
| $E_1(0,0,0)$ | + | + | Non-existent | *No* |
| $E_2(N_e,0,0)$ | + | − | $\alpha_{ed} > 1, \alpha_{ec} > 1$ | *Yes* |
| $E_3(0,N_d,0)$ | + | − | $\alpha_{de} > 1, \alpha_{dc} > 1$ | *Yes* |
| $E_4(0,0,N_c)$ | + | − | $\alpha_{ce} > 1, \alpha_{cd} > 1$ | *Yes* |
| $E_5\left(0, \frac{(\alpha_{cd}-1)N_d}{\alpha_{dc}\alpha_{cd}-1}, \frac{(\alpha_{dc}-1)N_c}{\alpha_{dc}\alpha_{cd}-1}\right)$ | + | × | $\alpha_{dc}, \alpha_{cd} > 1$ 且 $\alpha_{de}, \alpha_{ce} < 0$ | *No* |
| $E_6\left(\frac{(\alpha_{ce}-1)N_e}{\alpha_{ec}\alpha_{ce}-1}, 0, \frac{(\alpha_{ec}-1)N_c}{\alpha_{ec}\alpha_{ce}-1}\right)$ | + | × | $\alpha_{ec}, \alpha_{ce} > 1$ 且 $\alpha_{ed}, \alpha_{cd} < 0$ | *No* |
| $E_7\left(\frac{(\alpha_{de}-1)N_e}{\alpha_{de}\alpha_{ed}-1}, \frac{(\alpha_{ed}-1)N_d}{\alpha_{de}\alpha_{ed}-1}, 0\right)$ | + | × | $\alpha_{ed}, \alpha_{de} > 1$ 且 $\alpha_{ec}, \alpha_{dc} < 0$ | *No* |
| $E_8(x^*, y^*, z^*)$ | × | − | $\begin{cases} 1 - \frac{2y_e^*}{N_e} - \alpha_{de}\frac{y_d^*}{N_d} - \alpha_{ce}\frac{y_c^*}{N_c} < 0 \\[1mm] 1 - \frac{2y_d^*}{N_d} - \alpha_{ed}\frac{y_e^*}{N_e} - \alpha_{cd}\frac{y_c^*}{N_c} < 0 \\[1mm] 1 - \frac{2y_c^*}{N_c} - \alpha_{ec}\frac{y_e^*}{N_e} - \alpha_{dc}\frac{y_d^*}{N_d} < 0 \end{cases}$ | *No* |

population themselves achieve maximum value; When $\alpha_{de} > 1$, $\alpha_{dc} > 1$, the connection between digital platforms, and core enterprise and university research institutions remains competitive synergy, albeit within a contentious competitive framework. Digital platforms that leverage their unique competitive advantages impede the growth of the other populations. When $\alpha_{ce} > 1$, $\alpha_{cd} > 1$, while competing for resources, university research institutions suppress the other two populations, and ultimately reduce the scale of the core enterprise and the digital platform populations to zero. During the evolution of digital innovation ecosystem synergy, these three situations will impede the expected growth of specific populations, making it challenging for various groups to exchange resources and collectively benefit from each other. These will inhibit the construction and development of ecosystems.

In the case that it is difficult to accurately obtain a large amount of time-series data on digital innovation ecosystems, empirical research cannot be easily performed [58,59]. Then, numerical simulation is an effective method to explore the internal symbiotic evolution law [4,6]. However, numerical simulation methods have certain limitations in that they do not comprehensively reflect real scenarios. Considering this gap, and following existing theoretical frameworks and relevant studies [3,22,25], this study utilizes the real-life case of the innovation ecosystem of Hengyang City's Hengzhou Digital Economy Corridor to validate the symbiotic evolution mechanism of the digital innovation ecosystem.

## Numerical simulation

After analyzing the stability of the system strategy, we assign specific values for the growth rates of the different clusters, as well as the exogenous variables in the model. Following the literature on the configuration of symbiotic evolutionary parameters of innovative ecosystems [23], we set the natural growth rates of core enterprise, digital platform, and university research institute populations as 0.05, 0.01, and 0.02, respectively. The maximum size of these populations under constrained system resources is 1000, and their initial size is 100. Symbiosis is anticipated to evolve throughout 1000 units of time. By integrating Eq (1) with the pertinent data from Table 1, MATLAB 2017a is used to model the symbiotic evolutionary principles of distinct groups by combining various symbiosis elements.

**Freestanding symbiotic mode.**   When all the symbiotic factors are set to zero, the clusters within the digital innovation ecosystem display an autonomous synergistic pattern. The results of the evolutionary simulations are shown in Fig 2. The three groups do not engage with each other, and each develops independently. Their respective natural growth rates determine the rate of population growth. Over time, the three clusters reach a state of equilibrium and eventually their extraordinary sizes through independent development.

**Competitive symbiosis mode.**   Under the competitive symbiosis mode, the three types of digital innovation populations compete for resources that promote innovation. This competition relies on the changing values of the synergy component and can be divided into two types.

(1) Equal competition model

When the symbiosis coefficient deviates from equality and falls within the range of (0,1), the evolutionary simulation yields the outcomes depicted in the equal competition mode, as illustrated in Fig 3A and 3B. In this model, the core enterprise populations compete for innovative resources equally with the other two groups. The pace of development is influenced by their natural growth rates as well as the populations of digital platforms and university research institutes, which undergo a process of rapid growth followed by a gradual decline. Moreover, when the number of digital platforms surpasses that of university research institutions, the

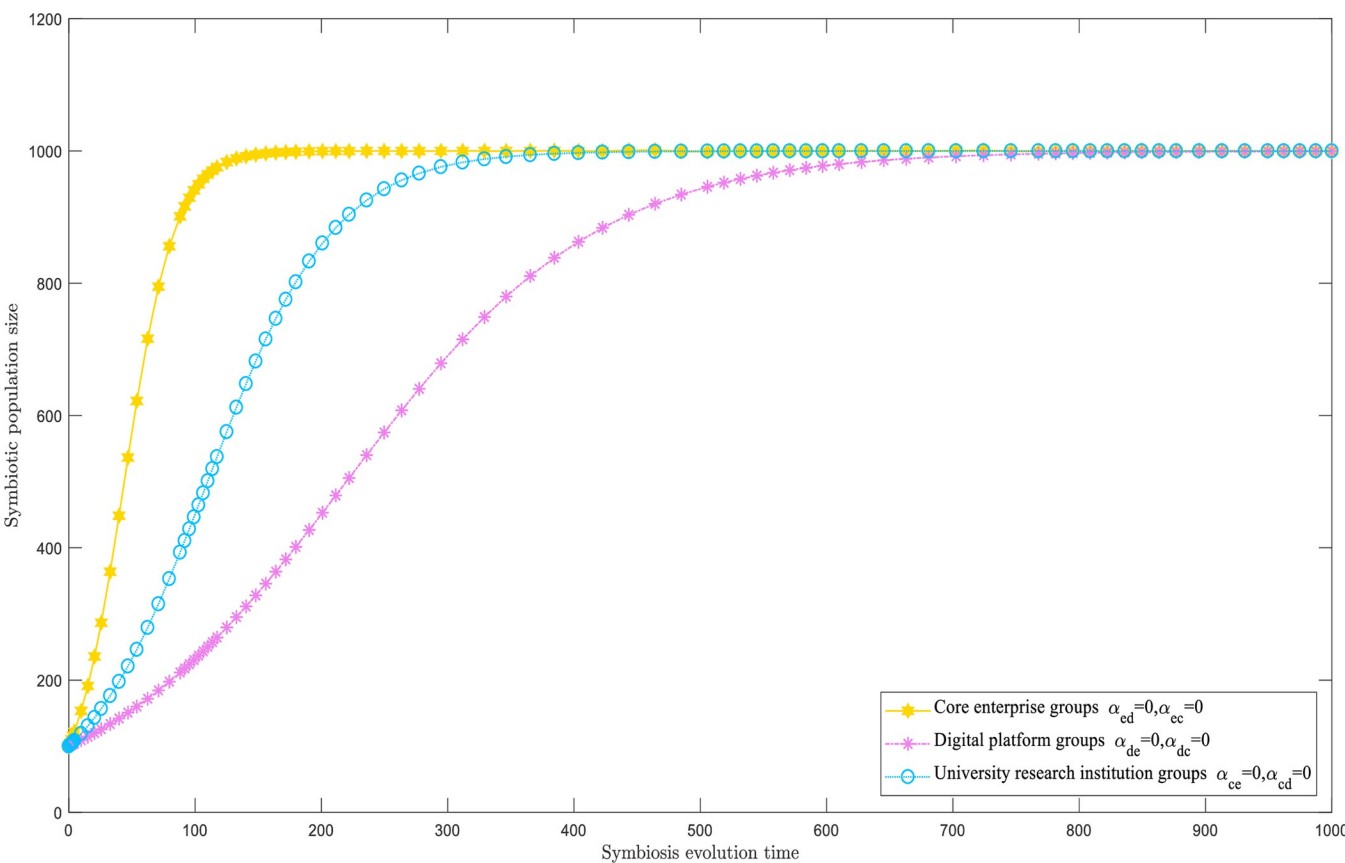

**Fig 2. Freestanding symbiotic model.**

growth rate becomes significantly slower, resulting in $\alpha_{ed} < \alpha_{ec}$ is obtained. Over time, the number of digital platforms will inevitably surpass that of university science institutions, thus granting them a predominant position in the core enterprise population. When the core enterprise population primarily determines the number of digital platforms, we get $\alpha_{ed} < \alpha_{ec}$. The population of university research institutes acquires a more powerful competitive edge than that of digital platforms. Moreover, the former will exceed the latter.

(2) Malignant competition mode

When the symbiosis coefficient is asymmetric and arbitrary one-sided symbiosis approaches one, it signals a malignant competitive pattern. The evolutionary simulation findings can be observed in Fig 3C and 3D. When the number of individuals in the core business exceeds the number of individuals on a digital platform, the mutual benefit discovered from the population of university research institutes, university platforms, and university scientific institutions is deployed by the core business population to utilize a noteworthy number of digital innovation resources. Nevertheless, this population decreases after a brief development period, whereas the core enterprise population grows and eventually becomes the largest. When the core population of enterprises transitions to the digital platform population, the symbiotic association with the population of university research institutes diminishes over time. Consequently, the core enterprise population fails to sustains its advantages, thus losing the ability to effectively compete for resources. This allows the other two populations to

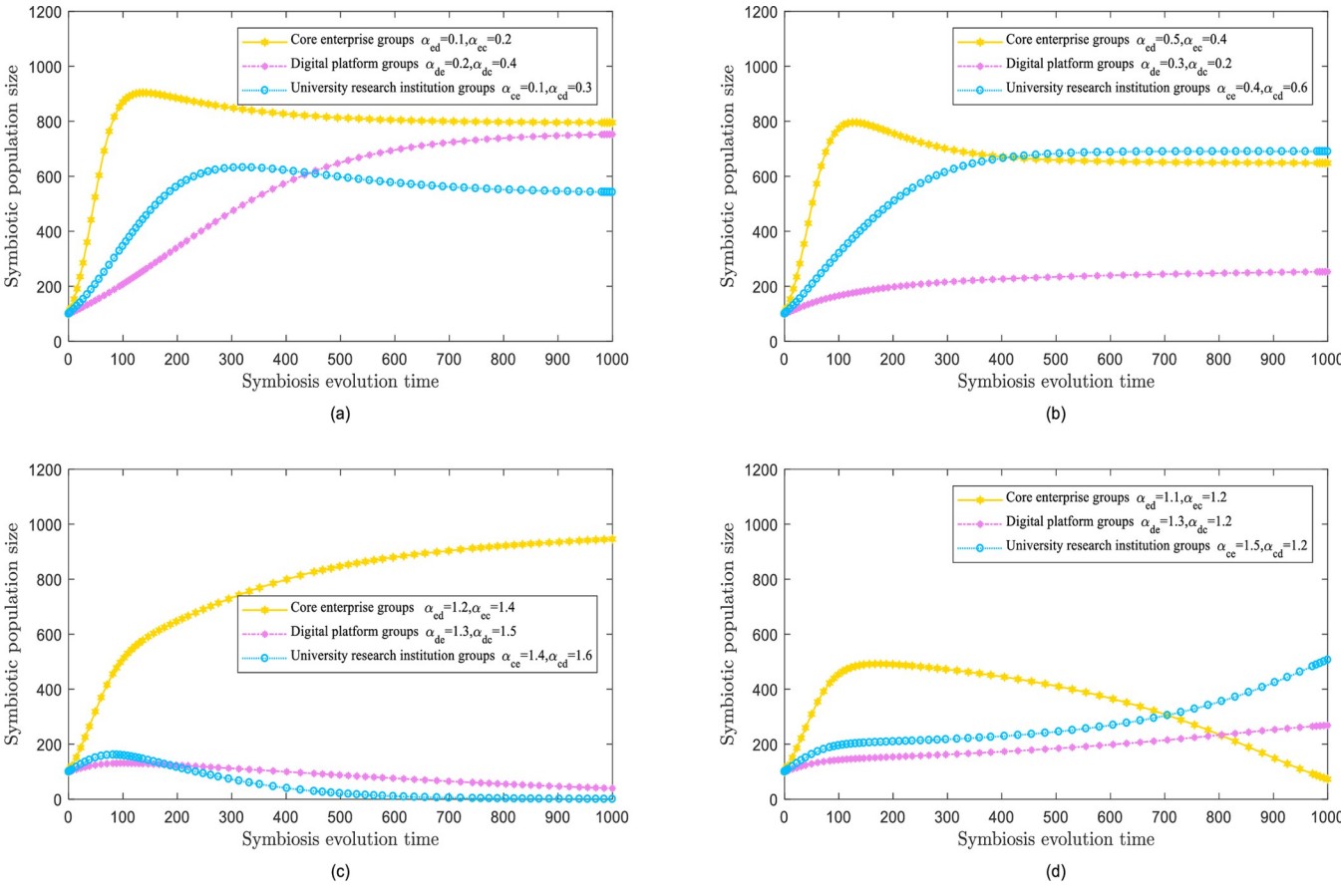

**Fig 3. Competitive symbiosis model.**

develop and finally beat the core enterprise population. Thus, the core enterprise's population faces a brief period of growth following by a natural decline, ultimately giving up its competitive status and advantages.

**Parasitic symbiotic mode.** In the parasitic symbiosis mode, the symbiotic factor between the two groups is mutually exclusive: negative (positive) symbioses are favorable (detrimental) to the population. Moreover, numerous scenarios can be generated based on different values of the symbiosis factor.

Fig 4A illustrates the findings of the evolutionary simulation when digital platform and university research institute populations interact with core enterprise populations. The proliferation of digital platforms and university research institutes is depleting the resources of core enterprises and triggering a decline in their growth after a brief increase. Although the final population size has not reached its maximum capacity, digital platforms and university research institutes have thrived by exploiting the resources of core enterprises. In particular, the population size of university science institutes exceeds that of other populations.

Next, Fig 4B illustrates the findings of the evolutionary simulation when core enterprise populations and university research institutes interact with digital platform populations. The digital platform is parasitized upon by the other populations, reducing its population growth and preventing it from achieving its maximum potential. Meanwhile, core enterprises and university research institutions receive resources through this parasitic relationship, experiencing

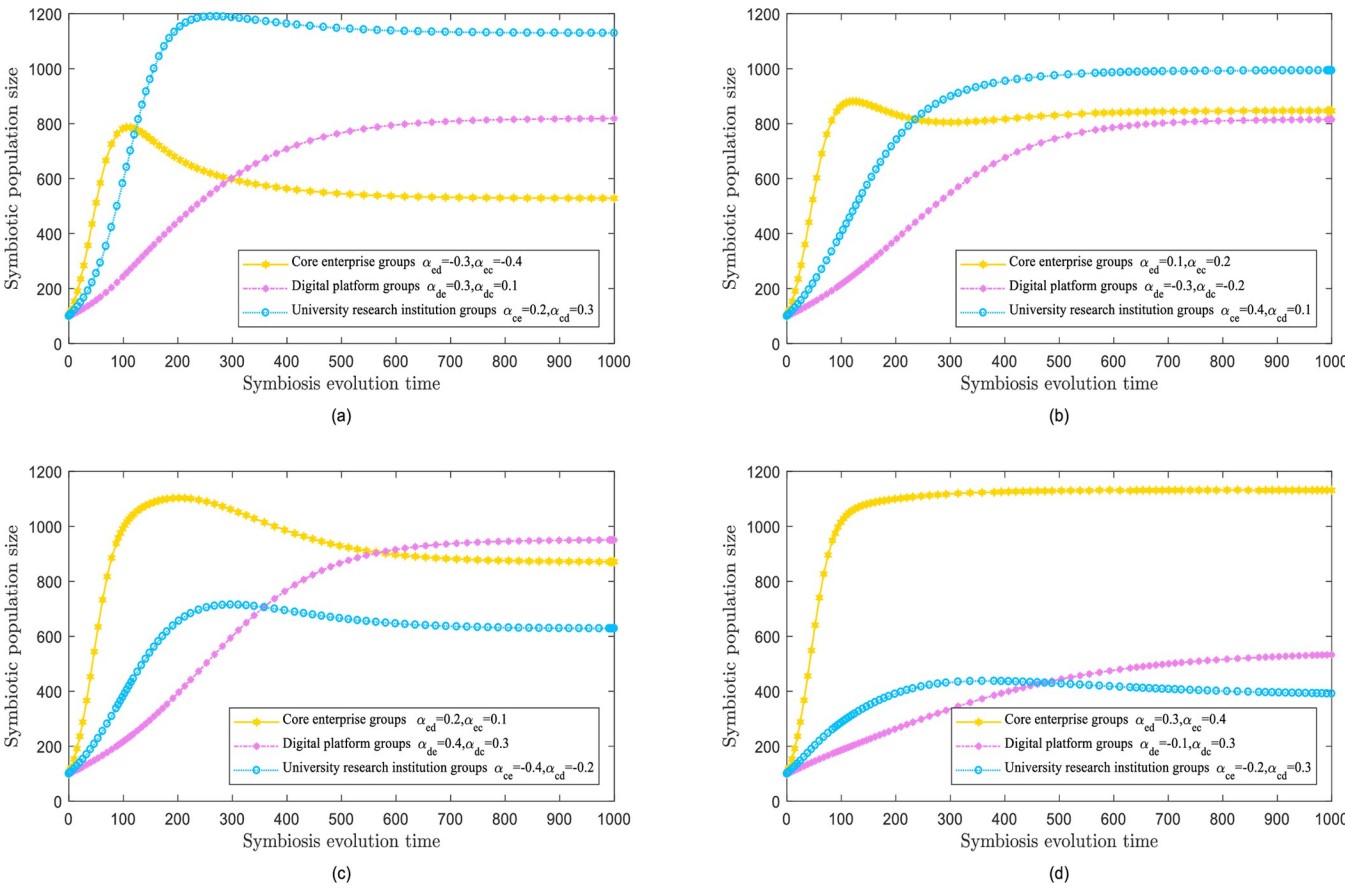

**Fig 4. Parasitic symbiotic mode.**

better growth. Crucially, university scientific institutions can accomplish their ultimate population size.

Fig 4C illustrates the findings of the evolutionary simulation when the core enterprise and digital platform populations coexist with the university research institute population. Due to the ongoing exhaustion of resources by core enterprise and digital platform population, the university population first jumps temporarily but eventually declines to a suboptimal population level.

When the core populations of enterprises parasitize digital platforms and university research

institutes, the findings of evolutionary simulation are shown in Fig 4D. Core enterprise populations benefit enormously from parasitism, experiencing crucial advancements in their development momentum and reach the population size boundaries. In contrast, the populations of digital platforms and college research institutions devour substantial resources, demonstrating slow progress, and lag behind.

**Biased mutual benefit mode.** In biased mutual benefit mode, the symbiotic coefficient is enhanced by hostile populations, whereas populations with zero symbiosis remain unaffected. The various values of symbiotic coefficients generate numerous scenarios.

The results of the evolution simulation are shown in Fig 5A. The core enterprise populations prosper from symbiotic interactions, whereas the digital platform populations and university research institutes have limited influence. The core enterprise population has gone

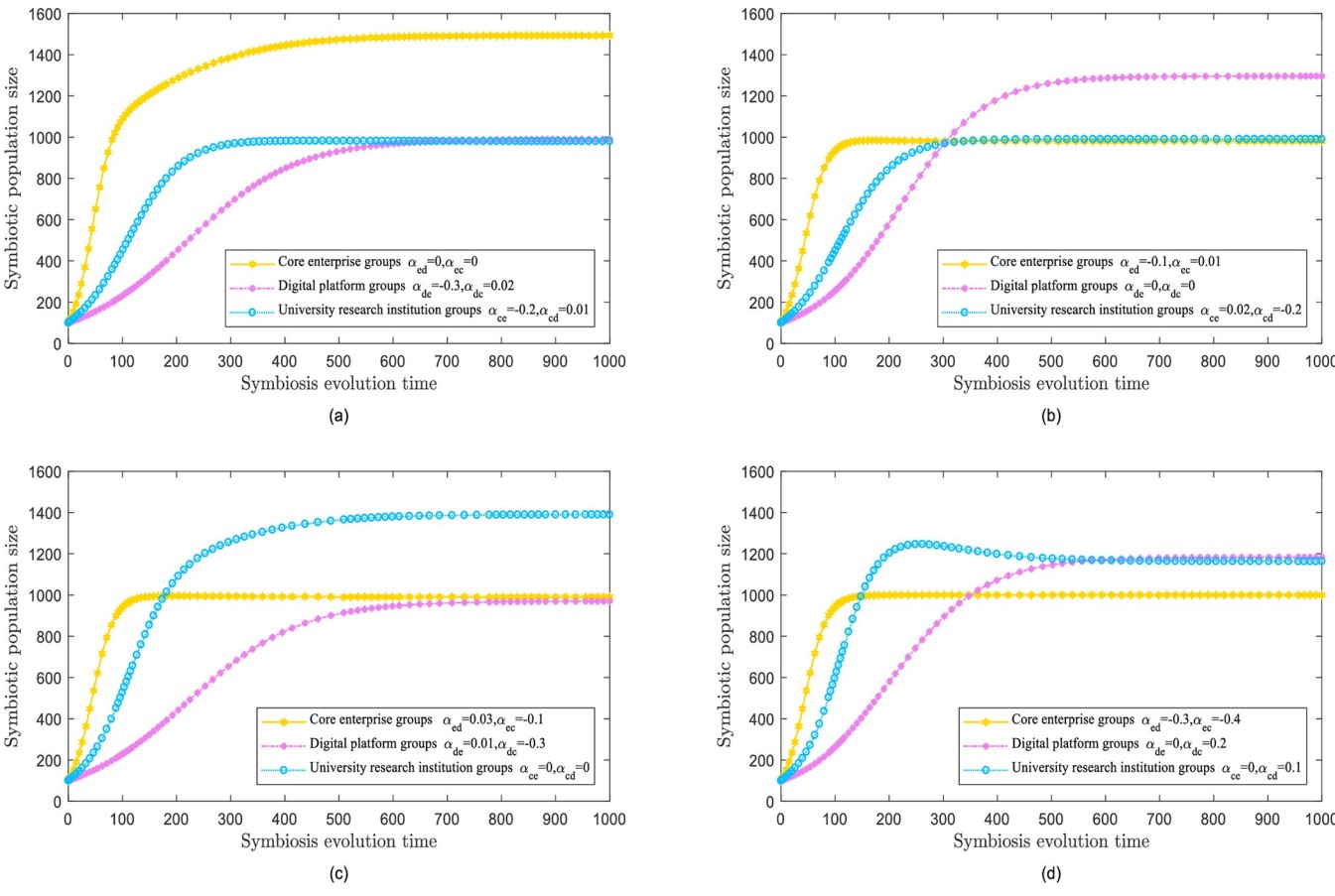

**Fig 5. Biased mutual benefit mode.**

through vital development owing to elevated resource availability. The population growth ceiling $N_e$ exceeded 1000 and reached 1400. The population of digital platforms and university research institutes, which are not considered influencers in the preferential symbiosis relationship, are affected by their natural growth rate, and their respective average growth and population growth ceilings $N_d$, $N_c$, which remain constant.

Next, Fig 5B shows that the digital platform population thrives in a symbiotic relationship, whereas the core enterprise population and university research institutes do not have any influence. The population of digital platforms has expanded in response to preferential symbiosis. The population growth $N_d$ has exceeded 1000 and achieved 1300. Surprisingly, the core enterprise population and the population at the university research institute, who are not directly involved in this symbiotic relationships have experienced regular growth, and their population ceiling $N_e$, $N_c$ has continued to be identical.

The evolution simulation findings depicted in Fig 5C imply that the population of university research institutes assists symbiotic relationships. In contrast, the core enterprise and digital platform populations exhibit no impact. As a recipient of the favorable symbiotic relationship, the population of university research institutes has been continuously growing, which is attributed to the constant influx of resources. The constraint on development is decreasing, and the maximum limit of population growth $N_c$ has exceeded 1000, obtaining 1400. The populations of the core enterprise and digital platforms, undisturbed by symbiotic

relationships, continue to grow at their standard rates, with their respective growth roof $N_e$, $N_d$ staying unmodified.

The results of the evolution simulation shown in Fig 5D show that the digital platform populations and university research institutions prosper from a symbiotic relationship, although core enterprise populations are not influenced. The core enterprise population, unaffected by external factors, typically advances with respect to its natural growth rate as long as it reaches its maximum population size by employing independent development. Conversely, the population of digital platforms and university research institutes confronts upsides, and experiecnces imnimumminimize obstacles to development, resulting in a rapid growth trend. The population growth ceilings $N_d$, $N_c$ exceeded 1000, accomplishing 1200 by the end.

**Mutually beneficial symbiotic model.** When all symbiosis parameters are unfavorable, populations of digital innovation ecosystems engage in mutually positive synergies through evolutionary simulations, as illustrated in Fig 6. The three types of populations reveal synergistic and mutually beneficial relationships. The upper limit of each population's size has increased to a certain extent, with the increase being molded by the symbiosis factor $\boldsymbol{\alpha}$. The magnitude of $\boldsymbol{\alpha}$ immediately correlates with the time of the rise in the population's size ceiling. Specifically, the people of the digital platform previously experienced significant increases in their size ceilings when $\boldsymbol{\alpha}_{ed}$ and $\boldsymbol{\alpha}_{cd}$ had larger absolute values. The maximum ceiling size for the digital platform population approaches 1700.

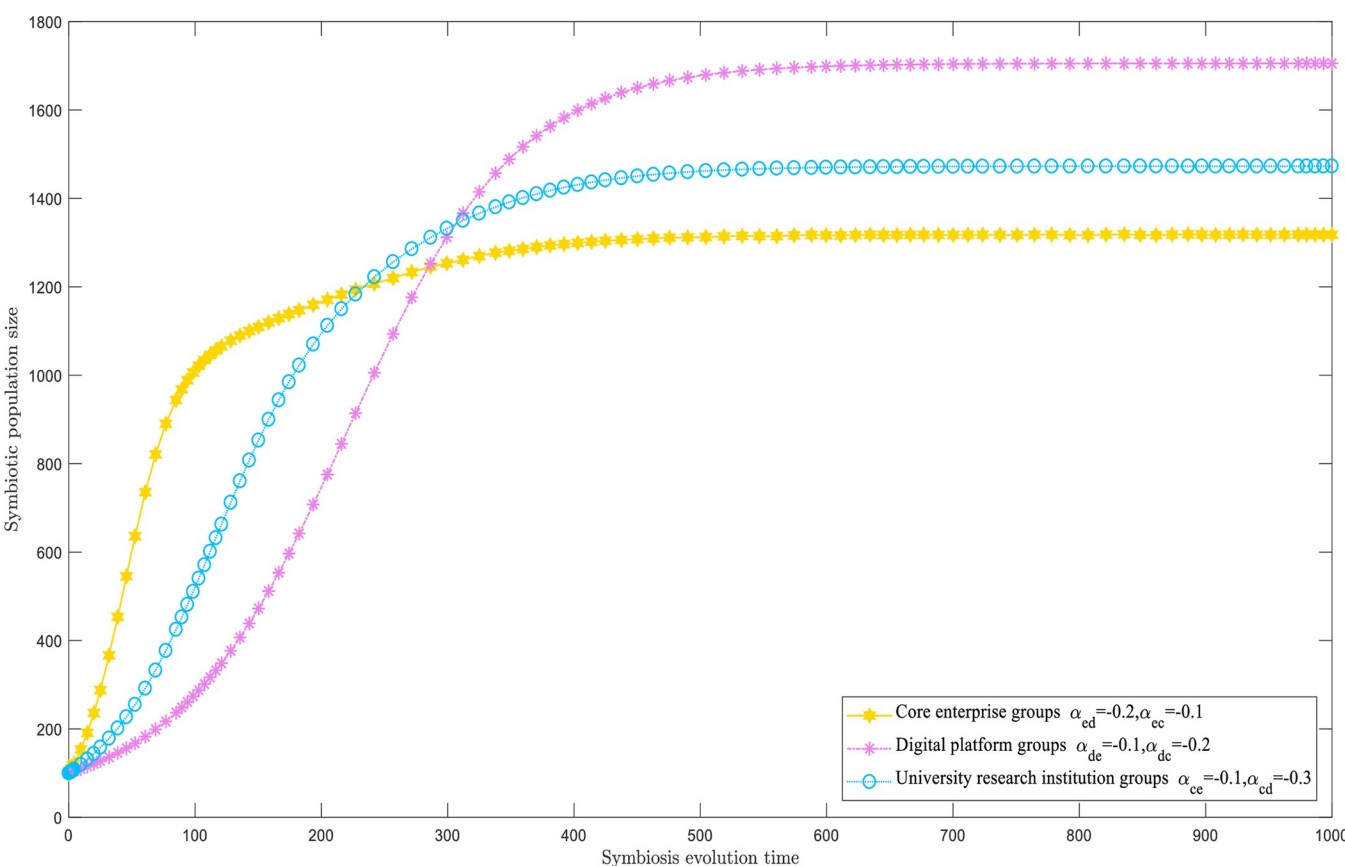

**Fig 6. Mutually beneficial symbiotic model.**

## Case analysis

Introducing actual cases can effectively supplement the inadequacy of numerical simulation. In our case it can help unveil the "black box" of the symbiotic evolution path of digital innovation ecosystems and make the research results more in line with reality. The Hengyang City Hengzhou Avenue Digital Economy Corridor Innovation Ecosystem has two main lines along Hengzhou Avenue. It has a total investment of 10 billion yuan and spans a length of 23 km. The corridor covers a radiation area of 50 square kilometers and is the primary infrastructure of Hengyang City's digital economy. It is designed to be a "golden corridor" for the digital economy featuring multiple parks and chains. The main structure comprises a single belt, numerous parks, and various chains. "One belt" refers to using Hengzhou Avenue, commencing from the Hengyang East High-Speed Railway Station in the eastern direction and concluding at the Yumu Campus of the University of South China on the western border. This route encompasses the Zhuhui District, Yanfeng District, Steamboat District, High-tech District, and Hengnan County. "Multiple parks" denotes the arrangement of 45 industrial parks that encompass the digital economy sectors such as mobile internet, blockchain, software, information technology, etc. Finally, "Multi-chain" signifies the effort to facilitate the convergence of the innovation, industrial, talent, capital, and policy chains. This endeavor aims to establish a top-notch digital industrial ecosystem and foster a significant advantage in advancing the digital economy of Hengyang.

The Digital Economy Corridor is an innovative ecosystem that employs wolf pack tactics. It has brought together over 30 prominent digital economy core enterprises, including Huawei, ZTE, Tencent, Boss Software, KDDI, Optical Cloud Technology, and Hikvision. Additionally, it has attracted 860 expanding core enterprises and fostered "invisible champions" in specialized fields like RAMON Technology, Tahoe Telecom, Northern Optoelectronics, and TBEA. The area contains 104 innovation platforms, of which over 40 industry platforms for the digital economy been established. These platforms include "Smart Valley," "Cloud Valley," and "Artificial Intelligence Base." In addition to hosting industry-university-research cooperation among many renowned universities, such as Shanghai Jiao Tong University and BeiHang University, these first-class industry-bearing digital platforms, comparable to the coasts have also brought together a large number of leading growing core enterprises such as China Electronics, Sinosoft International, Boss Software, and Jetco Digital. The corridor has also brought together eight universities, such as the University of South China, Hengyang Normal University, Hunan Institute of Technology, and research institutes, such as Shenzhen Foresight Industry Research Institute.

The symbiotic evolutionary mechanism of core enterprises, digital platforms, and university
research institutes in the digital innovation ecosystem is as follows:

First, the primary businesses contribute constructively and engage in profitable synergistic interactions to facilitate the advancement of the industrial chain. The company growth strategy known as the "incubator-industrial park-training center" is designed around a central enterprise as its guiding force. Several prominent digital firms, including Huawei, ZTE, and Tencent, have played a pivotal role in fostering the growth of various prospective small and medium-sized enterprises in the digital economic corridor of Hengzhou Avenue. The digital economy firms can be further helped by exploring the potential of platforms that incorporate mobile Internet, blockchain, 5G semiconductor, and e-commerce industrial parks. The Hengyang Enterprise Listing Counseling Center has been developed as a premier platform in the province to offer comprehensive and integrated counseling and development services for digital enterprises seeking listing themselves in capital markets. By implementing listings, the

center strives to strengthen the understanding of modern enterprise systems and capital markets among digital enterprises in the city. The ultimate goal is to enable the prosperous listings of extraordinary digital enterprises, and create a mutually beneficial and win-win cultivation system for the modern digital industry.

Second, digital platforms foster favorable and mutually advantageous symbiotic relationships in advancing financial and policy networks. Leveraging digital platforms, such as Industrial Wealth Link, Hengyang Smart Valley, China Telecom, China Unicom, and other similar platforms, along with digital arithmetic technologies, such as low-latency networks, massive storage, and high-performance computing, this initiative aims to integrate resources from various actors, including the enterprises, universities, and research institutes. The primary objective is to facilitate the connection between enterprises and social investment institutions, and foster strategic collaboration. Consequently, this collaboration seeks to enhance the provision of comprehensive digital financial services throughout the lifecycle of enterprises, thereby mitigating capital constraints. In addition, the organization offers a virtual platform that facilitates expedited growth in the digital financial service sector. Simultaneously, providing a virtual platform simplifies the expeditious development of digital government, and improvements in industrial, talent, financial, competition, and green policies. This, in turn, stimulates the advancement of scientific and technological achievements and innovative ideas within the digital realm, ultimately establishing a prominent hub for policy formulation in the digital economy.

Third, university research institutes play an ever-changing and beneficial symbiotic role in nurturing talent development and allowing innovation chains to progress. The Digital Economy Corridor benefits from the presence of approximately 180,000 college teachers and students. Moreover, local colleges, universities, and professional colleges, such as the Software College, Industrial Internet College, and Zhu Rong Live College, have established disciplines and specialties related to the digital economy. These academic institutions contribute significantly to the intellectual resources for the development of digital platforms within the corridor. Meanwhile, the implementation of initiatives such as "Ten Thousand Geese in Hengyang," "Leading Goose," and "Full Sky Star" has effectively drawn in a substantial number of entrepreneurial leaders, and top-tier scientific and technological experts in the digital economy. Consequently, nearly 20000 young talents have been pulled to Hengyang for innovation and entrepreneurship. This flurry of intellectual innovation elements has transformed Hengyang into a thriving hub for scientific and technological innovation, investment, and entrepreneurial activities within the region.

Thus, the digital innovation ecosystem comprises symbiotic units that collaborate on sharing new data, technology, and information generated by the symbiotic evolution system within the digital corridor. This promotes a mutually beneficial symbiotic mode, which leads to four separate symbiotic effects between symbiotic units within a specific digital economic environment: synergistic, information transmission, technology spillover, and cultural proliferation effects. The synergistic effect pertains to collaboration and mutual beneficial development of related entities, resulting in an optimal overall outcome within the synergistic evolutionary system. The information transmission effect refers to providing a virtual platform by digital technologies for exchanging information among these synergistic units. The technology spillover effect signifies the engagement of enterprise groups within the system in innovative endeavors, facilitating technological advancements among other synergistic units. Finally, the cultural proliferation effect denotes establishing a culture that values mutual benefit and win-win situations, thus fostering a positive role in disseminating such cultural values.

## Discussion

Here, we discuss the numerical simulation results. First, under the independent symbiosis mode, various populations successively reach their maximum size in their respective equilibrium states over time. Second, under the competitive symbiosis mode, the size of the coefficient of symbiosis between the core enterprise population and other populations determines the final competitive outcomes of various populations. Third, under the parasitic symbiosis mode, populations acting as parasites can obtain more significant benefits and develop rapidly, whereas the parasitized populations are hindered in their development owing to restrictions. Fourth, under the biased symbiosis mode, populations that are beneficiaries of the biased symbiotic relationship develop better. By contrast, non-influential populations grow independently and reach the upper limit of their sizes. Fifth, under the mutualistic symbiosis mode, populations work together to achieve mutual benefits. These numerical simulation results are consistent with existing symbiosis theories and the theoretical framework constructed in the previous section.

Based on the numerical simulation analysis, the conclusions of the case study analysis are as follows: First, the symbiotic evolution mechanism of the innovation ecosystem of Hengyang City's Hengzhou Avenue Digital Economy Corridor is that each core enterprise, digital platform, and university research institute is in a mutually beneficial symbiosis mode in a specific economic environment and produces four symbiotic effects. Second, core enterprises provide opportunities for the development of digital platforms and university research institutes in the process of promoting the development of the digital industry. Third, digital platforms provide funds to core enterprises and accelerate the construction of digital government to encourage the development of university research institutes. Fourth, university research institutes convey innovative digital talent for core enterprises and digital platforms. The above conclusions infuse the simulation results with practical implications that communities of interest within the digital innovation system need to share data, information, and digital technologies to enhance digital innovation collaboration for mutual benefit.

The simulation results and case conclusions are consistent with existing theories. The conclusions drawn from our selected actual cases are in line with innovation ecosystem theory [3]. The value co-creation between core enterprises and other communities of interest, such as digital platforms and university research institutes, plays an important role in the sustainable and healthy development of digital innovation ecosystems. Compared with prior works [23,33], this study combines digital innovation and innovation ecosystem, and shows that the numerical simulation results of digital innovation ecosystem are consistent with the symbiosis theory and empirical research. Crucially, the mutually beneficial symbiotic relationship is the most desirable evolutionary direction of the digital innovation ecosystem. Efforts should be made to transform the symbiosis relationships between populations into the mutually beneficial symbiosis relationship.

## Conclusion

We construct a theoretical framework based on symbiosis theory, and use numerical simulations and case study methods to study the symbiosis evolution mechanism of the digital innovation ecosystem. We establish a dynamic evolution model for the symbiosis of multiple groups in a digital innovation ecosystem with core enterprises, digital platforms, and university research institutes. Then, we perform the stability analysis of game strategies, undertake simulations of different symbiosis modes, and perform verification via actual cases. The main conclusions are as follows:

First, the digital innovation ecosystem is a complex adaptive system comprising three main components: core enterprise, digital platform, and university research institution. These symbiotic units engage in symbiotic activities, such as value co-creation within their respective populations. The specific symbiotic relationships formed by various symbiotic modes in particular symbiotic environments determine the nature of these activities. The digital innovation ecosystem demonstrates a diverse symbiotic nature, interactive synergistic nature, digital modularity, dynamic equilibrium, and other characteristics associated with symbiotic evolutionary structures. Second, the outcome of the symbiotic evolution in digital innovation ecosystems is controlled by symbiotic interactions that arise from various combinations of values for symbiotic interaction coefficients between populations. Third, the optimal evolutionary trajectory for digital innovation ecosystems involves cultivating a mutually advantageous symbiotic association. Enhancing the symbiotic relationship between populations is imperative, thereby facilitating the more efficient advancement of digital innovation ecosystems.

## Managerial implication

Compared with previous studies that used quantitative reasoning, the theoretical contributions of this study are as follows: First, compared with some studies [6,32], this study introduces the Logistic population growth model in ecological theory and combines it with evolutionary game theory to construct a symbiotic dynamic evolution model of the digital innovation ecosystem of core enterprises, digital platforms, and university research institutes. It explores the impact of the symbiotic evolutionary behavior of the three parties on the digital innovation ecosystem, providing a new direction for the application of evolutionary game theory. Second, advancing extant research [23,33], this study uses two complementary research methods, numerical simulations, and case studies, to enrich the literature on the symbiotic evolution mechanism of digital innovation ecosystems, and deepen the theory related to digital innovation ecosystems.

## Practical implications

These findings have the following implications for promoting mutually beneficial symbiotic relationships within the digital innovation ecosystem: First, core enterprises need to capitalize on their dominant advantages and embrace the collaborative framework of "industry-university-research-use-government" to systematically investigate the new development paradigm in the digital economy era. They should actively assimilate digital innovation resources within the system and bolster their core competitiveness in digital innovation. Second, university research institutions should utilize digital platforms to speed up data and information exchanges with core enterprises. This collaboration will enable continuous knowledge innovation, research, and development of new technologies, and cultivation of proficient digital technical professionals. Consequently, the talent and industrial chains should be integrated. Third, there is a need to expedite the establishment of various national-level dual innovation platforms, incubators, academic workstations, post-doctoral entrepreneurship stations, and other digital platforms. Additionally, increasing collaboration with university research institutes is crucial for encouraging mutual trust, effective communication, and the secure sharing of educational information. These efforts aim to advance digital talent training systems in alignment with contemporary demands. The government should expand its oversight of digital platforms, consistently improve their value co-creation capabilities, promote the establishment of fair and mutually beneficial cultural systems, and dissuade opportunistic actions that impede the digital innovation of other entities.

This study also has some limitations. First, we have only considered three subjects in the digital innovation ecosystem. However, multiple symbiotic subjects actually exist, and can affect its symbiotic evolution mechanisms. Future studies should consider other subjects, such as the government. Second, the numerical simulation of the symbiotic evolution mechanism is based on limited data. Future works can consider mining data to have a better sample size for simulation. Finally, this research presents a case study but does not elaborate on how to apply digital technology to the case for drawing more accurate conclusions; scholars should examine the important role of digital technology [60].

## Supporting information

**S1 File.**
(PDF)

## Author Contributions

**Conceptualization:** Liping Wu.

**Data curation:** Liping Wu.

**Formal analysis:** Liping Wu.

**Funding acquisition:** Yuqiong Li.

**Investigation:** Yuqiong Li.

**Methodology:** Liping Wu.

**Project administration:** Yuqiong Li.

**Resources:** Yuqiong Li.

**Software:** Liping Wu.

**Supervision:** Yuqiong Li.

**Validation:** Liping Wu.

**Visualization:** Liping Wu.

**Writing – original draft:** Liping Wu.

**Writing – review & editing:** Yuqiong Li.

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
