## [Decision Letter · Decision Letter 0]

27 Dec 2023

PONE-D-23-38488Simulation study on symbiotic evolution of digital innovation ecosystemsPLOS ONE

Dear Dr. Wu,

Thank you for submitting your manuscript to PLOS ONE. After careful consideration, we feel that it has merit but does not fully meet PLOS ONE’s publication criteria as it currently stands. Therefore, we invite you to submit a revised version of the manuscript that addresses the points raised during the review process.

We look forward to receiving your revised manuscript.

Best regards,

Prof. (Assist.) Donato Morea, Ph.D.

Academic Editor

PLOS ONE

5. Please provide a complete Data Availability Statement in the submission form, ensuring you include all necessary access information or a reason for why you are unable to make your data freely accessible. If your research concerns only data provided within your submission, please write "All data are in the manuscript and/or supporting information files" as your Data Availability Statement.

6. Please upload a copy of Figure 1-6. If the figure is no longer to be included as part of the submission please remove all reference to it within the text.

Reviewers' comments:

Reviewer's Responses to Questions

**Comments to the Author**

1. Is the manuscript technically sound, and do the data support the conclusions?

Reviewer #1: Partly

Reviewer #2: Yes

2. Has the statistical analysis been performed appropriately and rigorously? 

Reviewer #1: Yes

Reviewer #2: No

3. Have the authors made all data underlying the findings in their manuscript fully available?

Reviewer #1: No

Reviewer #2: Yes

4. Is the manuscript presented in an intelligible fashion and written in standard English?

Reviewer #1: Yes

Reviewer #2: Yes

5. Review Comments to the Author

Reviewer #1: The reviewer believes that the topic “Simulation study on symbiotic evolution of digital innovation ecosystems” is worthy of investigation. However, the following needs to be addressed. There are minor and major issues that should be corrected. I believe the paper could be further strengthened by added information about.

1.The title does not provide a core theme of the topic.

2.Please specify the source of the simulation data.

3.The language of this manuscript needs help from native speakers.

4.Please underscore the scientific value-added to your paper in your abstract. Your abstract should clearly state the essence of the problem you are addressing, what you did and what you found and recommend. That would help a prospective reader of the abstract to decide if they wish to read the entire article.

5.Introduction. This a very vague statement. These sentences do not provide any information on how the concept could be conceptualized? - The Introduction should have 1) a concise but complete justification of the topic's importance both academically and practically, and 2) an explanation of the gaps both in research and practice. Please review appropriate literature in the Introduction, with the research question clearly arising from that review.

6.What authors wanted to convey. Here author must build research gap following the previous studies.-The manuscript does not answer the following concerns: Why is it timeliness to explore such a study? What makes this study different from the previously published studies? Are there any similarly findings in line with the previously published studies? Are the findings different from prior academic studies that were conducted elsewhere, if any? What it requires, what are the new technologies, some recent issue highlights the importance. See the following: New Energy-Driven Construction Industry: Energy development in rural China towards clean energy system: Utilization status, co-benefit mechanism and countermeasures. Frontiers in Energy Research..

7.-There is no flow in the text. It partly depends on the lack of proofreading but also on the fact that many statements and claims are made without being followed up by a clear and logical discussion. It is especially problematic in the Introduction that brings up a number of findings from different areas without linking them together.

8.-More importantly, the choice of the questionnaire questions should be explained in light of the theory and the prior literature on the topic. The arguments are simply relationships and causes very close to the replication of many studies dealing with the same thing. For example, what is connection of upgrading path of manufacturing enterprises from the value perspective. See the following: Developing a conceptual partner matching framework for digital green innovation of agricultural high-end equipment manufacturing system toward agriculture 5.0: A Novel Niche Field Model Combined With Fuzzy VIKOR. Frontiers in Psychology, 13, 924109.

9.-Methodology: Model.. I suggest authors here build your main heading on Research and data methodology. Clearly explain the model building process, and what previous studies have used similar models (model testing approach).

See the following: "Incentive Mechanism for the Development of Rural New Energy Industry: New Energy Enterprise–Village Collective Linkages considering the Quantum Entanglement and Benefit Relationship", International Journal of Energy Research. https://doi.org/10.1155/2023/1675858.

A stochastic differential game of low carbon technology sharing in collaborative innovation system of superior enterprises and inferior enterprises under uncertain environment, https://doi.org/10.1515/math-2018-0056

10.The authors should emphasize the important role of digital technology in future research. See the following: The Interaction Mechanism and Dynamic Evolution of Digital Green Innovation in the Integrated Green Building Supply Chain. Systems 2023, 11, 122. https://doi.org/10.3390/systems11030122.

11.Please consider this structure for manuscript final part.

-Discussion

-Conclusion

-Managerial Implication

-Practical/Social Implications

12.Please make sure your conclusions' section underscores the scientific value-added of your paper, and/or the applicability of your findings/results. Highlight the novelty of your study. In addition to summarizing the actions taken and results, please strengthen the explanation of their significance. It is recommended to use quantitative reasoning comparing with appropriate benchmarks, especially those stemming from previous work.

Reviewer #2: The introduction effectively outlines the focus on the symbiotic evolution of digital innovation ecosystems, setting the stage for the simulation study. However, there is a need for more explicit articulation of the research objectives and the specific research questions that the simulation aims to answer. Providing this clarity would enhance the reader's understanding of the study's purpose.

The paper briefly mentions the use of simulation but lacks a thorough justification for this methodology. Why was simulation chosen over other research methods? A more explicit discussion on the advantages and limitations of simulation in the context of studying digital innovation ecosystems would strengthen the research design.Simulation studies are heavily reliant on accurate validation and calibration processes. The paper should elaborate on how the simulation model's outputs were validated against real-world data or existing theoretical frameworks. A robust validation process enhances the reliability of simulation results.To connect innovation.See the following. Exploring the Effect of Buyer Engagement on Green Product Innovation: Empirical Evidence from Manufacturers. Business Strategy and the Environment. https://doi.org/10.1002/bse.2631

Interorganizational collaboration for innovation improvement in manufacturing: The mediating role of social performance, International journal of innovation management, https://doi.org/10.1142/S1363919620500498.

While the paper mentions the simulation study's key findings, a more detailed presentation and discussion of these results are needed. How do the observed patterns or outcomes align with existing theories or expectations? A thorough analysis of the simulation results would contribute to the paper's significance and impact.The paper could benefit from a discussion of the practical implications of the simulation findings. How can stakeholders in digital innovation ecosystems apply the insights derived from the simulation to enhance collaboration, innovation, or sustainability? This discussion will enhance the paper's relevance to practitioners and policymakers.The paper needs a more robust integration with existing literature on digital innovation ecosystems. How do the simulation findings align with or challenge existing theories or empirical studies in this field? A comparative analysis would provide a richer context for interpreting the simulation results.

The conclusion should include a section on future research directions. Identifying potential areas for further exploration based on the simulation study's limitations or unanswered questions would contribute to the ongoing scholarly conversation on digital innovation ecosystems.

The paper is generally well-written, but attention should be given to clarity and readability. Complex terms or concepts should be explained, and the logical flow of the paper should be enhanced for easier comprehension.In summary, while the paper addresses an intriguing topic, improvements in articulating research objectives, justifying the simulation methodology, detailing model development, validating and calibrating the simulation, presenting key findings, discussing practical implications, comparing with existing literature, suggesting future research directions, and enhancing clarity would significantly strengthen the overall quality of the manuscript.

6. PLOS authors have the option to publish the peer review history of their article (what does this mean?). If published, this will include your full peer review and any attached files.

Reviewer #1: No

Reviewer #2: No

---

## [Author Response · Author response to Decision Letter 0]

5 Feb 2024

Dear Editor and Reviewers:

 Thank you for your letter and the reviewers’ comments concerning our manuscript entitled “Simulation study on symbiotic evolution of digital innovation ecosystems” (ID: PONE-D-23-38488). Those comments and suggestions are all valuable and very helpful for revising and improving our paper, as well as the essential guiding significance to our research.

 We sincerely appreciate your recognition of our work and the suggestions you offered us, which play an essential role in improving the quality of the paper. We hope the revised manuscript can meet your requirements. 

Yours Sincerely,

Liping Wu

School of Economics Management and Law

University of South China 

Hengyang 421001 China 

E-mail: 19907344375@163.com

February 2, 2024

---

## [Decision Letter · Decision Letter 1]

26 Feb 2024

Multi-group symbiotic evolutionary mechanisms of a digital innovation ecosystem: Numerical simulation and case study

PONE-D-23-38488R1

Dear Dr. Wu,

We’re pleased to inform you that your manuscript has been judged scientifically suitable for publication and will be formally accepted for publication once it meets all outstanding technical requirements.

Kind regards,

Prof. Donato Morea, Ph.D.

Academic Editor

PLOS ONE

Reviewers' comments:

Reviewer's Responses to Questions

**Comments to the Author**

1. If the authors have adequately addressed your comments raised in a previous round of review and you feel that this manuscript is now acceptable for publication, you may indicate that here to bypass the “Comments to the Author” section, enter your conflict of interest statement in the “Confidential to Editor” section, and submit your "Accept" recommendation.

Reviewer #1: (No Response)

Reviewer #2: All comments have been addressed

2. Is the manuscript technically sound, and do the data support the conclusions?

Reviewer #1: (No Response)

Reviewer #2: Yes

3. Has the statistical analysis been performed appropriately and rigorously? 

Reviewer #1: (No Response)

Reviewer #2: Yes

4. Have the authors made all data underlying the findings in their manuscript fully available?

Reviewer #1: (No Response)

Reviewer #2: Yes

5. Is the manuscript presented in an intelligible fashion and written in standard English?

Reviewer #1: (No Response)

Reviewer #2: Yes

6. Review Comments to the Author

Reviewer #1: The manuscript has significantly improved as compared to the previous version. Indeed, the authors tried to improve it, and the main weaknesses are solved.

Thus, in my opinion, the manuscript is recommendable for publication..

Reviewer #2: I find the topic of your research highly interesting, and I appreciate the effort you have put into presenting your work in a clear and concise manner. Your research is valuable to our readership, and I believe it will make a meaningful contribution to the field. The overall quality of your work, along with the thoroughness of your revisions, has led to its acceptance. I commend your dedication to improving the manuscript and effectively addressing the reviewers' comments.

7. PLOS authors have the option to publish the peer review history of their article (what does this mean?). If published, this will include your full peer review and any attached files.

Reviewer #1: No

Reviewer #2: No

---

## [Editor Report · Acceptance letter]

27 Mar 2024

PONE-D-23-38488R1 

PLOS ONE

Dear Dr. Wu, 

I'm pleased to inform you that your manuscript has been deemed suitable for publication in PLOS ONE. Congratulations! Your manuscript is now being handed over to our production team.

Kind regards, 

on behalf of

Professor (Associate) Donato Morea 

Academic Editor

PLOS ONE